# Structured flexibility in recurrent neural networks via neuromodulation

**Julia C. Costacurta***
Stanford University
jcostac@stanford.edu

**Shaunak Bhandarkar***
Stanford University
shaunakb@stanford.edu

**David Zoltowski**
Stanford University
dzoltow@stanford.edu

**Scott W. Linderman**
Stanford University
scott.linderman@stanford.edu

## Abstract

A core aim in theoretical and systems neuroscience is to develop models that help us better understand biological intelligence. Such models range broadly in both complexity and biological plausibility. One widely-adopted example is task-optimized recurrent neural networks (RNNs), which have been used to generate hypotheses about how the brain's neural dynamics may organize to accomplish tasks. However, task-optimized RNNs typically have a fixed weight matrix representing the synaptic connectivity between neurons. From decades of neuroscience research, we know that synaptic weights are constantly changing, controlled in part by chemicals such as neuromodulators. In this work we explore the computational implications of synaptic gain scaling, a form of neuromodulation, using task-optimized low-rank RNNs. In our neuromodulated RNN (NM-RNN) model, a neuromodulatory subnetwork outputs a low-dimensional neuromodulatory signal that dynamically scales the low-rank recurrent weights of an output-generating RNN. In empirical experiments, we find that the structured flexibility in the NM-RNN allows it to both train and generalize with a higher degree of accuracy than low-rank RNNs on a set of canonical tasks. Additionally, via theoretical analyses we show how neuromodulatory gain scaling endows networks with gating mechanisms commonly found in artificial RNNs. We end by analyzing the low-rank dynamics of trained NM-RNNs, to show how task computations are distributed.

## 1 Introduction

Humans and animals show an innate ability to adapt and generalize their behavior across various environments and contexts. This suggests that the neural computations producing these behaviors must have flexible dynamics that are able to adjust to these novel conditions. Given the popularity of recurrent neural networks (RNNs) in studying such neural computations, a key question is whether this flexibility is (1) adequately and (2) accurately portrayed in RNN models of computation.

Traditional RNN models have fixed input, recurrent, and output weight matrices. Thus, the only way for an input to impact a dynamical computation is via the static input weight matrix. Prior work has shown that flexible, modular computation is possible with these models [1], but looking to the biology suggests alternative ways of modeling. In particular, experimental neuroscience research has shown that synaptic strengths in the brain (akin to weight matrix entries in RNNs) are constantly changing — in part due to the influence of neuromodulatory signals [2].

Neuromodulatory signals are powerful and prevalent influences on neural activity and subsequent behavior. Dopamine, a well-known example, is implicated in motor deficits resulting from Parkinson's disease and has been the subject of extensive study by neuroscientists [3]. For computational study,

38th Conference on Neural Information Processing Systems (NeurIPS 2024).

neuromodulators are especially interesting because of their effects on synaptic connections and learning [4]. In particular, neuromodulators have been shown to alter synaptic strength between neurons, effectively reconfiguring circuit dynamics [5].

In this work, we seek to incorporate a neuromodulatory signal into task-trained RNN models. We propose a model consisting of a pair of RNNs: a small "neuromodulatory" RNN and a larger, low-rank "output-generating" RNN. The neuromodulatory RNN controls the weights of the output-generating RNN by scaling each rank-1 component of its recurrent weight matrix. This allows the network to produce flexible, yet structured dynamics that unfold over the course of a task. We first review background work in both the machine learning and computational neuroscience literature. Next, we introduce the model and provide some intuition for the impact of neuromodulation on the model's dynamics, relating our model to the canonical long short-term memory (LSTM) network. We end by presenting the performance and generalization capabilities of neuromodulated RNNs on a variety of neuroscience and machine learning tasks, showcasing the ability of a relatively simple, biologically-motivated augmentation to enhance the capacity of RNN models.

## 2 Background

First, we review related work in the theoretical neuroscience and machine learning literature.

### 2.1 Modeling neuromodulatory signals

Our work builds on a body of literature dating back to the 1980s, when pioneering computational neuroscientists added neuromodulation to small biophysical circuit models (for reviews, see [6–8]). These models consist of coupled differential equations whose biophysical parameters are carefully specified to simulate biologically-accurate spiking activity. As Marder relates in her retrospective review [5], such neuromodulatory models were created in response to neuronal circuit models which viewed circuit dynamics as "hard-wired". Neuromodulatory mechanisms offered an answer to experimental observations that anatomically fixed biological circuits were capable of producing variable outputs [9, 10]. Of particular relevance to this work, in his 1990 paper Abbott [11] showed that adding a neuromodulatory parameter to an ODE model of spiking activity allows a network of neurons to display capacity for both long- and short-term memory and gate the learning process, anticipating the LSTMs that would become prominent a few years later. We also draw comparisons between our model and the LSTM in the sections that follow. However, our work does not aim to model any specific biophysical system; rather, it aims to begin to bridge the gap between these highly biologically-accurate models and general network models (i.e. RNNs) of neuronal activity by adding a biologically-motivated form of structured flexibility.

More recent attempts to model neuromodulation have taken advantage of increased computational power. Prior work has investigated modulatory influence in spiking neural networks, for example by linearly scaling the firing rates of a subset of neurons [12], incorporating arousal-mediated modulatory signals to induce phase transitions [13], and facilitating credit assignment during learning [14]. Stroud et al. [15] use a balanced excitatory/inhibitory RNN to model motor cortex, and incorporate modulatory signals as constant multipliers on each neuron's activity. Our work similarly proposes a modulatory signal that impacts network activity, but we instead allow this signal to scale factors of the recurrence matrix. Duong et al. [16] use a similar factor scaling approach to model adaptive whitening in early sensory processing. Our approach differs by focusing on low-rank recurrence matrices in a task-oriented setting. In addition to RNN models, gain modulation has been explored in neural ODEs [17] and feedforward networks [18].

The works of Tsuda et al. [19] and Vecoven et al. [20] are most similar to what we present here. In Tsuda et al. [19], the authors train RNNs using constant, multiplicative neuromodulatory signals applied to pre-specified subsets of the recurrent weights. They show that these neuromodulatory signals allow an otherwise fixed network to perform variations of a task. In contrast, we employ time-dependent neuromodulatory signals that allow dynamics to evolve throughout tasks. Instead of pre-specifying the values and regions of impact of neuromodulators, we allow the model to learn the time-varying neuromodulatory signal and what segment of the neuronal population it impacts. Vecoven et al. [20] use a two-network approach in which a neuromodulatory network processes contextual information to alter the activation functions of a "main" feedforward deep neural network. They show that this "Neuro-Modulated Network" outperforms RNNs on meta-RL benchmarks. We also use a two-network approach in which one network modulates the parameters of the other, but instead applied to recurrent neural networks in a non-RL paradigm.

## 2.2 Hypernetworks

Our approach is closely related to recent work using hypernetworks to enhance model capacity. Ha et al. [21] use small networks (termed hypernetworks) to generate parameters for layers of larger networks. In their HyperRNN, a hypernetwork generates the weight matrix of an RNN as the linear combination of a learned set of matrices. We also allow our neuromodulatory network to specify the weight matrix of a larger RNN as a linear combination of a learned set of matrices; however, our learned matrices are rank-1 to facilitate easier dynamical analysis and faster training. It is also worth noting that in practice, Ha et al. [21] simplify their HyperRNN so that the hypernetwork scales the rows of a learned weight matrix, which could be seen as postsynaptic scaling in our model.

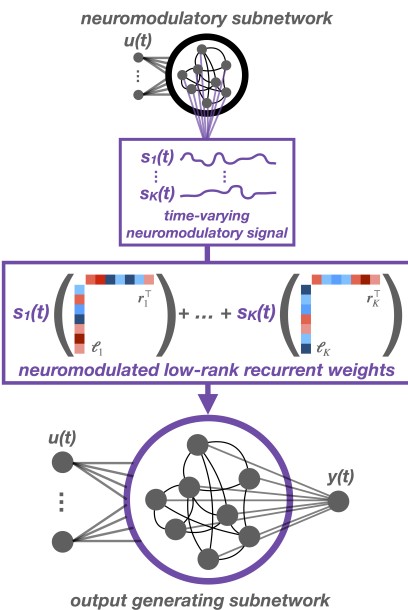

neuromodulatory subnetwork

time-varying neuromodulatory signal

neuromodulated low-rank recurrent weights

output generating subnetwork

Figure 1: The NM-RNN consists of two subnetworks: a low-rank subnetwork that generates the output (bottom) and a smaller, full-rank neuromodulatory subnetwork (top).

Similarly, von Oswald et al. [22] study the ability of hypernetworks to learn in the multitask and continual learning setting. They find that hypernetworks trained to produce task-specific weight realizations achieve high performance on continual learning benchmarks. In exploring potential neuroscience applications of their work, they remark that while their approach might be unrealistic, a hypernetwork that outputs lower-dimensional modulatory signals could assist in implementing task-specific mode-switching. We seek to obtain similar performance gains with a biologically plausible model of neuromodulation.

### 2.3 Low-rank recurrent neural networks

For a variety of tasks of interest, measured neural recordings are often well-described by a set of lower dimensional latent variables [23, 24] (although see [25, 26] for an alternative view). Likewise, artificial neural networks trained to solve tasks that mimic those found in neural experiments also often exhibit low-rank structure [27]. Based on these findings, recurrent neural networks with low-rank weight matrices (also called low-rank RNNs, LR-RNNs) have emerged as a popular class of models for studying neural dynamics [28–30]. Importantly, low-rank RNNs are able to model nonlinear dynamics which evolve in a low-rank subspace, offering potential for visualization and interpretation. Here, we leverage low-rank RNNs as ideal candidates for neuromodulation, with each factor of the low-rank recurrence matrix becoming a possible target for synaptic scaling.

## 3 Neuromodulated recurrent neural networks

Motivated by the appealing dynamical structure of low-rank RNNs and the ability of neuromodulation to add structured flexibility, we propose the *neuromodulated RNN* (NM-RNN). The NM-RNN consists of two linked subnetworks corresponding to neuromodulation and output generation. The output generating subnetwork is a low-rank RNN, which admits a natural way to implement neuromodulation. We allow the output of the neuromodulatory subnetwork to scale the low-rank factors of the output generating subnetwork's weight matrix. In particular, we propose incorporating neuromodulatory drive via a coupled ODE with neuromodulatory subnetwork state $\boldsymbol{z}(t) \in \mathbb{R}^M$ and output-generating state $\boldsymbol{x}(t) \in \mathbb{R}^N$:

$$\tau_z \frac{\mathrm{d}\boldsymbol{z}(t)}{\mathrm{d}t} = -\boldsymbol{z}(t) + \boldsymbol{W}_z\, \phi(\boldsymbol{z}(t)) + \boldsymbol{B}_z\boldsymbol{u}(t) \tag{1}$$

$$\tau_x \frac{\mathrm{d}\boldsymbol{x}(t)}{\mathrm{d}t} = -\boldsymbol{x}(t) + \boldsymbol{W}_x\big(\boldsymbol{z}(t)\big)\, \phi(\boldsymbol{x}(t)) + \boldsymbol{B}_x\boldsymbol{u}(t) \tag{2}$$

$$\boldsymbol{y}(t) = \boldsymbol{C}\boldsymbol{x}(t) + \boldsymbol{d}, \tag{3}$$

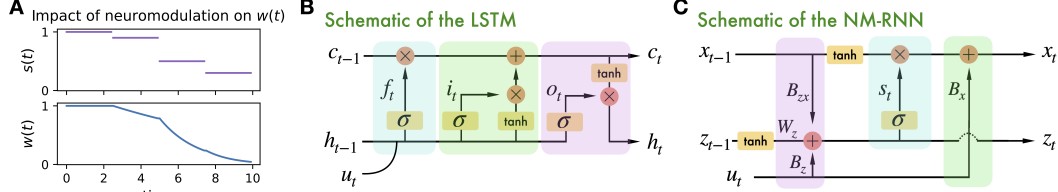

Figure 2: **A.** Illustration of how neuromodulatory signals $s(t)$ affect decay rates of the state $w(t)$ in a 1-D, simplified model (derived in Section 3.1), calculated with prespecified $s(t)$ signal. As $s(t)$ approaches 0, $w(t)$ decays more rapidly. **B. & C.** Visual comparison of an LSTM and an NM-RNN. Corresponding parts of the networks are highlighted with shaded rectangles. Blue: a forget gate computation. Green: an input gate to recurrent dynamics. Purple: recurrent feedback onto the modulatory state variable.

where the dynamics matrix $\boldsymbol{W}_x(\boldsymbol{z}(t))$ is a function of the neuromodulatory subnetwork state $\boldsymbol{z}(t)$ via a neuromodulatory signal $\boldsymbol{s}(\boldsymbol{z}(t)) \in \mathbb{R}^K$ which scales each low-rank component of $\boldsymbol{W}_x$:

$$\boldsymbol{s}(\boldsymbol{z}(t)) = \sigma(\boldsymbol{A}_z \boldsymbol{z}(t) + \boldsymbol{b}_z) \qquad \boldsymbol{W}_x(\boldsymbol{z}(t)) = \sum_{k=1}^{K} s_k(\boldsymbol{z}(t)) \, \boldsymbol{\ell}_k \boldsymbol{r}_k^\top. \qquad (4)$$

The output-generating subnetwork is modeled as a size-$N$ low-rank RNN where $\boldsymbol{u}(t) \in \mathbb{R}^P$ are the inputs, $\boldsymbol{B}_x \in \mathbb{R}^{N \times P}$ are the input weights, and $\phi(\cdot)$ is the tanh nonlinearity. The neuromodulatory subnetwork is modeled as a small vanilla RNN with its own time-constant $\tau_z$, recurrence weights $\boldsymbol{W}_z \in \mathbb{R}^{M \times M}$, and input weights $\boldsymbol{B}_z \in \mathbb{R}^{M \times P}$. To limit the capacity of the neuromodulatory subnetwork, we take its dimension $M$ to be smaller than the output-generating subnetwork's dimension $N$. We set $\tau_z \gg \tau_x$ since neuromodulatory signals are believed to evolve relatively slowly [31]. The neuromodulatory subnetwork state $\boldsymbol{z}(t)$ alters the rank-$K$ dynamics matrix $\boldsymbol{W}_x \in \mathbb{R}^{N \times N}$ via a $K$-dimensional linear readout $\boldsymbol{s}(\boldsymbol{z}(t))$, where $\sigma(\cdot)$ is the sigmoid nonlinearity. The components of $\boldsymbol{s}$ act as linear scaling factors on each rank-1 component $\boldsymbol{\ell}_k \boldsymbol{r}_k^\top \in \mathbb{R}^{M \times M}$ of $\boldsymbol{W}_x$. For ease of notation, in the rest of the text we write $\boldsymbol{s}(t)$ to mean $\boldsymbol{s}(\boldsymbol{z}(t))$. The output $\boldsymbol{y}(t) \in \mathbb{R}^O$ of the paired networks is a linear readout of $\boldsymbol{x}(t)$.

This augmentation to the RNN framework allows for structured flexibility in computation. In traditional RNNs, the recurrent weight matrix is fixed, and thus the inputs to the system can only perturb the state of the network. In the NM-RNN, the neuromodulatory subnetwork can use information from the inputs to dynamically up- and down-weight different low-rank components of the recurrent weight matrix, offering greater computational flexibility. As we will see below, this flexibility also allows the network to reuse dynamical components across different tasks and task conditions.

## 3.1 Mathematical intuition

To gain some intuition for the potential impacts of neuromodulation on RNN dynamics, first consider the case where $\boldsymbol{W}_x$ is symmetric (i.e., $\boldsymbol{\ell}_k = \boldsymbol{r}_k \, \forall k$), where $\{\boldsymbol{\ell}_k\}_{1 \le k \le K}$ forms an orthonormal set, where the nonlinearity is removed (i.e., $\phi(x) = x$), and where there are no inputs (i.e., $\boldsymbol{u}(t) = 0 \, \forall t$). We can then reparameterize the system with a new hidden state $\boldsymbol{w}(t) = \boldsymbol{L}^\top \boldsymbol{x}(t)$, where $\boldsymbol{L} \in \mathbb{R}^{N \times K}$ is the matrix whose columns are $\boldsymbol{\ell}_k$ (so that $\boldsymbol{L}^T \boldsymbol{L} = I$). This produces decoupled dynamics:

$$\tau_x \frac{\mathrm{d}\boldsymbol{w}(t)}{\mathrm{d}t} = -\boldsymbol{w}(t) + \mathbf{S}(t)\boldsymbol{w}(t) \qquad (5)$$

where $\boldsymbol{S}(t) = \mathrm{diag}(\boldsymbol{s}(t))$. Solving this ODE gives an equation for the components of $\boldsymbol{w}(t)$:

$$w_k(t) = w_k(0) \exp\left(-\int_0^t \frac{(1 - s_k(t'))}{\tau_x} \, \mathrm{d}t'\right)$$

From this equation and the visualization in Fig. 2A, we see that the decay rate of each component $w_k(t)$ is governed by its corresponding neuromodulatory signal $s_k(t)$. In this way, $\boldsymbol{s}(t)$ can effectively speed up or slow down decay of dynamic modes, similar to gating in an LSTM.

## 3.2 Connection to LSTMs

Having observed that neuromodulation can alter the timescales of dynamics, note further that the low-rank update for the linearized NM-RNN in eq. (5) mirrors the cell-state update equation for a long short-term memory (LSTM) cell [32]. Specifically, the neuromodulatory signal $s(t)$ resembles the forget gate of an LSTM (fig. 2B). Indeed, as in eq. (5), if we linearize the output-generating subnetwork of the NM-RNN and assume that $\boldsymbol{L} = \boldsymbol{R}$ (so that $\boldsymbol{W}_x$ is symmetric) and $\boldsymbol{L}^T \boldsymbol{L} = I$, then for $\boldsymbol{w}(t) = \boldsymbol{L}^T \boldsymbol{x}(t)$ and $\tau_x = 1$, the discretized low-rank dynamics are given by

$$\boldsymbol{w}_t = \boldsymbol{s}_t \odot \boldsymbol{w}_{t-1} + \boldsymbol{L}^T \boldsymbol{B}_x \boldsymbol{u}_t \tag{6}$$

LSTMs have two states that recurrently update across each timestep $t$: a hidden state $\boldsymbol{h}_t \in \mathbb{R}^{N_{\text{LSTM}}}$ and a cell state $\boldsymbol{c}_t \in \mathbb{R}^{N_{\text{LSTM}}}$. Equation (6) closely mirrors the cell-state update of the LSTM:

$$\boldsymbol{c}_t = \mathbf{f}_t \odot \boldsymbol{c}_{t-1} + \boldsymbol{i}_t \odot \tilde{\boldsymbol{c}}_t \tag{7}$$

Here, the forget gate $\mathbf{f}_t$ is a form of modulation that depends on the LSTM hidden state $\boldsymbol{h}_t$, much like the NM-RNN's neuromodulatory signal $\boldsymbol{s}(t)$ is a form of modulation depending on the NM-RNN's neuromodulatory subnetwork state $\boldsymbol{z}(t)$. The second term $\boldsymbol{i}_t \odot \tilde{\boldsymbol{c}}_t$ can be viewed as a gated transformation of the input signal $\boldsymbol{u}(t)$. In fact, under suitable assumptions, we show that the dynamics of an NM-RNN can be reproduced by those of an LSTM (see Supplementary Material, Proposition 1).

As a gated analog of the RNN, the LSTM has enjoyed greater success than ordinary RNNs in performing tasks that involve keeping track of long-distance dependencies in the input signal [33]. Thus, highlighting the connection between the NM-RNN and LSTM suggests the NM-RNN's ability to successfully model long-timescale dependencies, unlike regular RNNs (see Section 6).

## 4 Time interval reproduction

To evaluate the potential of neuromodulated RNNs to add structured flexibility, we first consider a timing task since neuromodulators such as dopamine are implicated in time measurement and perception [34]. In the Measure-Wait-Go (MWG) task (Fig. 3A) [35], the network receives a 3-channel input containing the measure, wait, and go cues. The network must measure the interval between the measure and wait cues, and reproduce it at the go cue by outputting a linear ramp of the same duration. Tasks such as this one are commonly used to study the neural underpinnings of timing perception in humans and non-human primates [36, 37].

### 4.1 Experiment matches theory for rank-1 networks

To investigate how the NM-RNN's neuromodulatory signal is constrained by the task requirements, we first consider a class of analytically tractable NM-RNNs: networks for which the output-generating subnetwork is linear (i.e., the tanh nonlinearity is replaced with the identity function) and rank-1. In this case, there is one pair of row and column factors, $\boldsymbol{\ell}$ and $\boldsymbol{r}$, respectively. If the target output signal is given by $f(t)$ and there are no inputs, then an NM-RNN that successfully produces this output signal will precisely have the neuromodulatory signal,

$$\boldsymbol{s}(t) = \frac{f(t) + \tau_x f'(t) - \boldsymbol{d}}{(\boldsymbol{\ell}^\top \boldsymbol{r})\left(f(t) - \boldsymbol{c}^T \boldsymbol{w}^\perp(0)e^{-t/\tau_x} - \boldsymbol{d}\right) + (\boldsymbol{c}^\top \boldsymbol{\ell})\left(\boldsymbol{r}^\top \boldsymbol{w}^\perp(0)e^{-t/\tau_x}\right)},$$

where our readout is $y = \boldsymbol{c}^\top \boldsymbol{x} + \boldsymbol{d}$ and $\boldsymbol{w}^\perp(t) := \boldsymbol{x}(t) - \frac{1}{||\boldsymbol{\ell}||_2^2}\boldsymbol{\ell}\boldsymbol{\ell}^\top \boldsymbol{x}(t)$ is the component of $\boldsymbol{x}(t)$ evolving outside of the column space of $\boldsymbol{\ell}$ (viewed as a matrix in $\mathbb{R}^{N \times 1}$). For the full derivation, see the Supplementary Material, Section B. If we assume further that $\boldsymbol{w}^\perp(0)$ is sufficiently small and $\tau_x$ is also sufficiently small, then we may make the approximation,

$$\boldsymbol{s}(t) \approx \frac{f(t) + \tau_x f'(t) - \boldsymbol{d}}{\boldsymbol{\ell}^\top \boldsymbol{r}\left(f(t) - \boldsymbol{d}\right)} = \frac{1}{\boldsymbol{\ell}^T \boldsymbol{r}}\left(1 + \frac{\tau_x f'(t)}{f(t) - \boldsymbol{d}}\right). \tag{8}$$

In the MWG task, no inputs are provided to the network during the ramping period following the go cue, so eq. (8) applies. In fig. 3B, we show that the neuromodulatory signal of a trained rank-1 NM-RNN during the ramping period matches closely with the theoretical prediction made by eq. (8) for both trained and extrapolated target intervals.

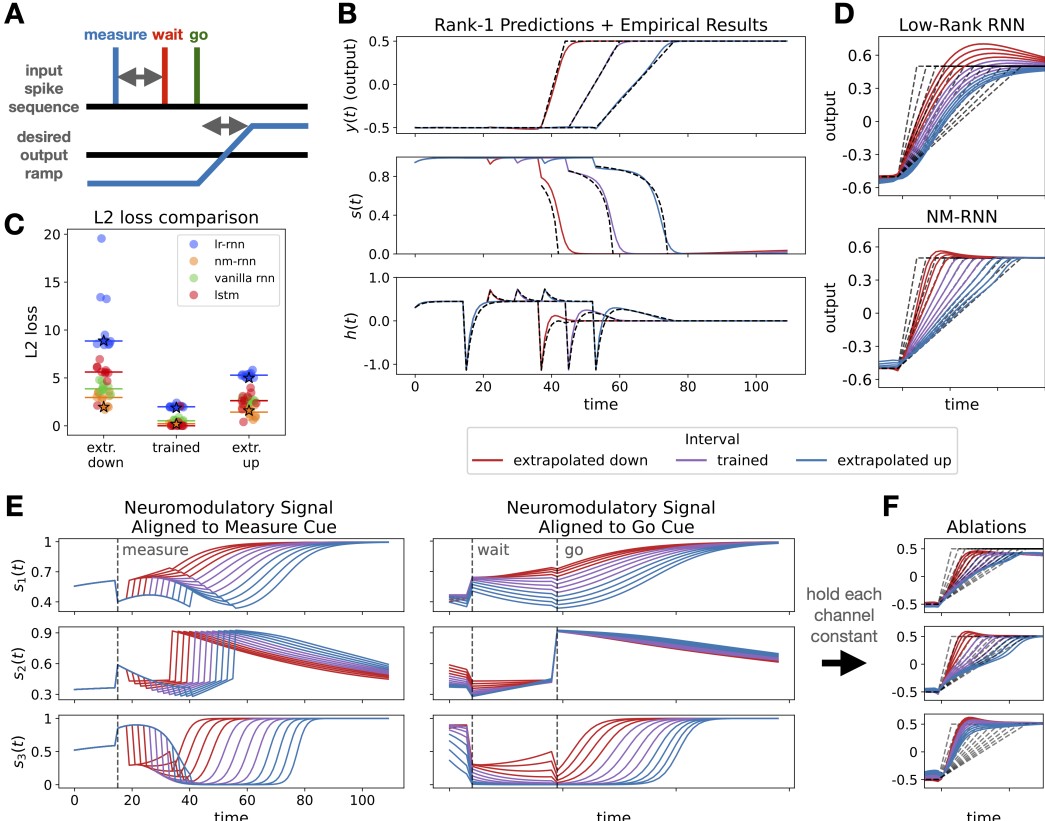

Figure 3: **A.** Visualization of Measure-Wait-Go task. **B.** Theoretical predictions (dashed lines) match closely with empirical results (solid lines) for rank-1 network. **C.** L2 loss comparison of four model types on MWG task. We trained 10 randomly initialized parameter-matched low-rank RNNs, NM-RNNs, vanilla RNNs, and LSTMs. Median losses shown by bars. Performance of visualized LR-RNN and NM-RNN are starred (example visualizations for all four models shown in Supp. Fig. 1). **D.** Comparison of model-generated output ramps for both trained (purple) and extrapolated (red, blue) intervals. Dashed lines show target outputs. **E.**Three-dimensional neuromodulatory signal $\boldsymbol{s}(t)$ for trained/extrapolated intervals. Left, traces are aligned to start of trial. Right, traces are aligned to 'go' cue. **F.** Resulting output traces when ablating each component of $\boldsymbol{s}(t)$. In all panels, colors reflect trained/extrapolated intervals (see legend). For output plots, dashed grey lines are targets. Additional model visualizations in Supp. Fig. 1-2.

## 4.2 Improved generalization and interpretability on timing task for rank-3 networks

To continue our analysis of NM-RNNs on the MWG task, we increase the rank of the output-generating subnetwork to three. We do this to compare to the networks shown in Beiran et al. [35] and to showcase networks with more degrees of structured flexibility. In Beiran et al. [35], the authors show that rank-3 low-rank RNNs perform and extrapolate better on this task when provided a tonic context-dependent input, which varies depending on the length of the desired interval. As we have mentioned, such sensory inputs to the network may only alter the resulting dynamics by being passed through the input weight matrix. We propose the NM-RNN as an alternative mechanism by which inputs may alter the network dynamics.

We trained parameter-matched NM-RNNs ($N = 100$, $M = 5$, $K = 3$, $\tau_x = 10$, $\tau_z = 100$), LR-RNNs ($N = 106$, $K = 3$, $\tau = 10$), vanilla RNNs ($N = 31$, $\tau = 10$), and LSTMs ($N = 15$) to reproduce four intervals, then tested their extrapolation to longer and shorter intervals. In the LR-RNN and NM-RNN, the low-rank matrix was chosen to have rank 3, as in Beiran et al. [35]. In fig. 3C, we plot the L2 losses for ten instances of each model. We see that although the vanilla RNN, LSTM, and NM-RNN are all able to train accurately, the NM-RNN consistently achieves a lower loss on the extrapolated intervals. In fig. 3D, we then show outputs for a typical LR-RNN and NM-RNN

(performance of these visualized networks indicated by stars in fig. 3C, visualizations of vanilla RNN and LSTM shown in Supp. Fig. 1). The outputs of the NM-RNN have more accurate slope and shape for both trained and extrapolated intervals, with the LR-RNN failing to reproduce shorter and longer extrapolated intervals.

We then investigate how the neuromodulatory signal $s(t)$ contributes to the task computation. Figure 3E shows the three dimensions of $s(t)$ plotted over the full range of trained and extrapolated intervals. Each dimension shows activity correlated to particular stages of the task. In fig. 3E (right), we see that $s_1(t)$ and $s_3(t)$ have activity highly correlated to the measured interval. In particular, we can see that between the wait and go cues, $s_1(t)$ separates shorter intervals from longer ones, setting up initial dynamics for the go period when the ramp is generated. The third component $s_3(t)$ appears to be involved with ending the output ramp, since it saturates first for the shorter intervals and then the longer ones. Figure 3F shows the result of ablating each dimension of $s(t)$ by keeping that component fixed around its initial value. We see that performance suffers in all cases, especially when ablating the effect of $s_1(t)$ and $s_3(t)$. Most dramatically, ablating $s_3(t)$ destroys the ability of the network to change the slope of the output ramp appropriately. These results show that the network uses its neuromodulatory signal to process timing information and generalize across task conditions.

## 5    Reusing dynamics for multitask learning

Next, we move beyond generalization within a single task to investigate the capabilites of the NM-RNN when switching between tasks. There has been recent interest in studying how neural network models might reassemble learned dynamical motifs to accomplish multiple tasks [1, 38]. Driscoll et al [1] showed that an RNN trained to perform an array of tasks shares modular dynamical motifs across task periods and between tasks. With this result in mind, we were curious how the NM-RNN might use its neuromodulatory signal to flexibly reconfigure dynamics across tasks.

We performed our analysis using the four-task set from Duncker et al. [39], which includes the tasks DelayPro, DelayAnti, MemoryPro, and MemoryAnti illustrated in fig. 4A. In the DelayPro task, the network receives a three-channel input consisting of a fixation input and two sensory inputs which encode an angle $\theta \in [0, 2\pi)$ as $(\sin(\theta), \cos(\theta))$. The fixation input starts and remains at 1, then drops to 0 to signal the start of the *readout* period, when the network must generate its response. The sensory inputs appear after a delay, and persist throughout the trial. The goal of the network is to produce a three-channel output which reproduces the fixation and sensory inputs. In the MemoryPro task, the sensory inputs disappear before the readout period, requiring the network to store $\theta$. In the 'Anti' tasks, the networks must instead produce the opposite sensory outputs, $(\sin(\theta+\pi), \cos(\theta+\pi))$, during the readout period. The task context is fed in as an additional one-hot input. These tasks are analogous to variants of the center-out reaching task, which has been used to study the neural mechanisms of motion in non-human primates [40].

To study the potential of NM-RNNs to flexibly reconfigure dynamics to perform a new task, we only fed the contextual inputs to the neuromodulatory subnetwork, and not to the output-generating subnetwork. This required the model to reuse the output-generating subnetwork's weights when adding a new task. We trained an NM-RNN to perform the first three tasks in the set (DelayPro, DelayAnti, MemoryPro), then froze the weights of the output-generating subnetwork and retrained only the neuromodulatory subnetwork's weights on the fourth task, MemoryAnti. We compared this to retraining the input weights of LR-RNNs, vanilla RNNs, and LSTMs, to investigate two strategies of processing context.

We trained parameter-matched NM-RNNs ($N = 100$, $M = 20$, $K = 3$, $\tau_x = 10$, $\tau_z = 100$), LR-RNNs ($N = 100$, $K = 3$, $\tau = 10$), vanilla RNNs ($N = 18$, $\tau = 10$), and LSTMs ($N = 8$) in this training/retraining framework. Figure 4B shows performance of example networks on the trained and retrained tasks, using the percent correct metric from Driscoll et al. [1]. The NM-RNN matches the performance of the LSTM and higher-rank vanilla RNN, and considerably outperforms the LR-RNN with no modulation. This performance gain over LR-RNNs is not the result of retraining more parameters; in fact, due to the contrasting sizes of the neuromodulatory and low-rank subnetworks, the input weight matrix of the comparison LR-RNN contains more parameters than the entire neuromodulatory subnetwork, since it must process all inputs (context, sensory, and fixation). To see exactly how the recurrent dynamics were rearranged for this new task, we plotted the neuromodulatory signal of an example network for learned and extrapolated tasks in fig. 4C.

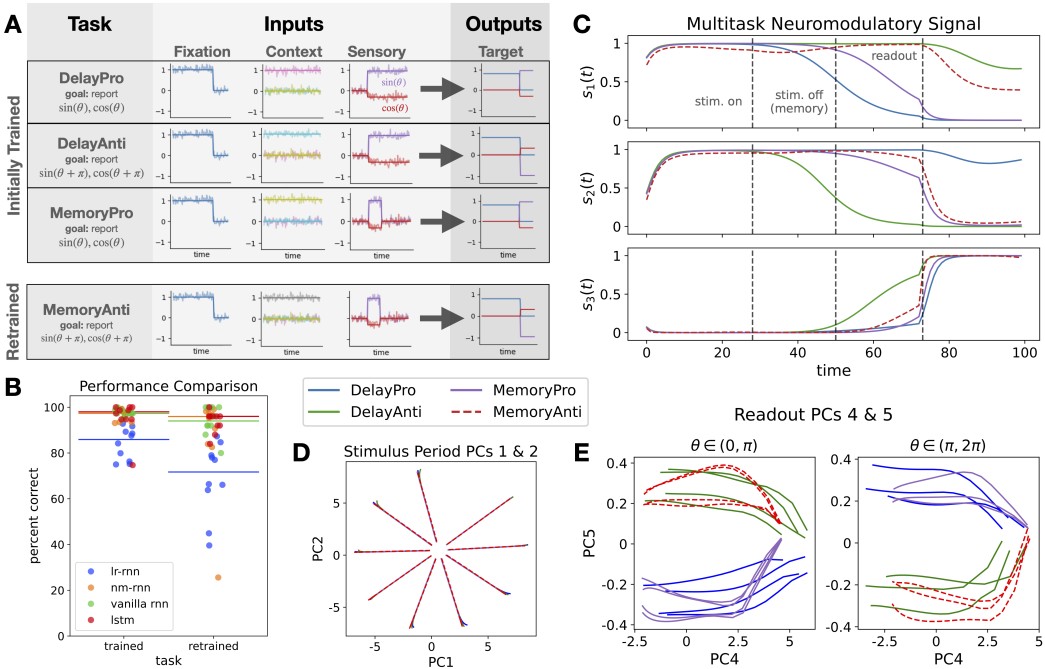

Figure 4: **A.** Depiction of inputs and targets for four tasks. Networks were trained on the first three tasks, then context-processing weights were retrained on the fourth task (final row). **B.** Performance comparison for four model types in multitask setting, for three initially trained tasks and retrained MemoryAnti task. Median performance shown by bars, with color of bar split in case of overlapping. We trained 10 randomly initialized parameter-matched low-rank RNNs, NM-RNNs, vanilla RNNs, and LSTMs. **C.** Neuromodulatory signal for an example network. **D.** Dynamical analysis of network activity during different stages of the tasks. (Left) PCs 1&2 during the stimulus period show a ring attractor which stores the measured angle. (Right) Sign of PC5 during readout period corresponds to Pro/Anti. Additional model visualizations in Supp. Fig. 3-4.

We then analyzed the dynamical structure of one of the NM-RNNs by performing PCA on the output-generating subnetwork's state $\boldsymbol{x}(t)$ for a variety of input angles. Figure 4D shows the first two PCs of the neural activity during the stimulus presentation period (before the stimulus shut off for Memory trials). During this period, the neural activity spreads out to arrange on a ring according to the measured angle. After the stimulus disappears in the MemoryPro/Anti tasks, the neural activity decays back along these axes, but it is still decodable based on its angle from the origin (see Supp. Fig. 4). To find how this model encoded Pro/Anti versions of tasks, we performed another PCA on the neural activity during the readout period. As shown in fig. 4E, the sign of PC5 during this period is correlated with whether the task is Pro or Anti. Curiously, the positive/negative relationship flips for $\theta \in (\pi, 2\pi)$, likely relating to the symmetric structure of sine and cosine. These results show the ability of the NM-RNN to flexibly reconfigure the dynamics of the output-generating subnetwork, both to solve multiple tasks simultaneously and to generalize to a novel task.

## 6 Capturing long-term dependencies via neuromodulation

Inspired by the similarity between the coupled NM-RNN and the LSTM (see section 3.2), we designed a sequence-related task with long-term dependencies, called the *Element Finder Task* (EFT). On this task, gated models like the NM-RNN outperform ordinary RNNs. When endowed with suitable feedback coupling from the output-generating subnetwork to the neuromodulatory subnetwork, the NM-RNN demonstrates LSTM-like performance on the EFT, while vanilla RNNs (with matched parameter count) fail to solve this task.

In the EFT (fig. 5A), the input stream consists of a query index, $q \in \{0, 1, \ldots, T-1\}$ followed by a sequence of $T$ randomly chosen integers. The goal of the model is to recover the value of the element at index $q$ from the sequence of integers. At each time for $t \geq 1$, the $t$th element of the sequence is

passed as a one-dimensional input to the model. At time $t = T$, the model must output the value of the element at index $q$ in the sequence. For our results (shown below), we took $T = 25$.

We trained several NM-RNNs, LR-RNNs, full-rank RNNs, and LSTMs on the EFT, conserving the total parameter count across networks. To emphasize its connection to the LSTM, each NM-RNN included an additional feedback coupling from $\boldsymbol{x}(t)$ to $\boldsymbol{z}(t)$:

$$\tau_z \frac{\mathrm{d}\boldsymbol{z}(t)}{\mathrm{d}t} = -\boldsymbol{z}(t) + \boldsymbol{W}_z \phi(\boldsymbol{z}(t)) + (\boldsymbol{B}_{zx}\,\phi(\boldsymbol{x}(t)) + \boldsymbol{b}_{zx}) + \boldsymbol{B}_z \boldsymbol{u}(t) \tag{9}$$

Each model used a linear readout with no readout bias. The resulting performances of each model tested are shown in fig. 5B. Figure 5C moreover illustrates the learning dynamics (as measured by MSE loss) for a single run of selected networks. Like LSTMs, NM-RNNs successfully perform the task, whereas low- and full-rank RNNs largely fail to do so.

To understand how a particular NM-RNN ($N = 18, M = 5, K = 8, \tau_x = 2, \tau_z = 10$) uses neuromodulatory gating to solve the EFT, we visualize the trial-averaged behavior of different components of $\boldsymbol{s}(t)$ across query indices ($q = 5, 10, 15, 20$), revealing that certain components of $\boldsymbol{s}(t)$ transition between 0 and 1 on a timescale correlated to the query index $q$ (fig. 5D; left and right); while other components zero out (fig. 5; middle). Visualizing a low-dimensional projection of $\boldsymbol{z}(t)$ across different query indices reveals that $\boldsymbol{z}(t)$ settles to a fixed point on an approximate line attractor encoding query index $q$ (fig. 5E). These findings show that $\boldsymbol{z}(t)$ attends to the query index, facilitating gate-switching behavior in $\boldsymbol{s}(t)$ upon arrival of the queried element.

Next, we analyze $\boldsymbol{x}(t)$ by visualizing its top 2 principal components across each combination of the query indices $q = 5, 10$, and $15$ and the target element values $-10, -5, 0, 5$, and $10$ (fig. 5F). Trajectories with different element values but the same query index start at the same location. Each trajectory converges towards the origin, and upon arrival of the query timestep, rapidly moves to a fixed point on an approximate line attractor that encodes element value. The arrangement of fixed points along this line moreover preserves the ordering of their corresponding element values. In summary, these results show that the NM-RNN solves the EFT by distributing its computations across the neuromodulatory subnetwork, which attends to the query index, and the output-generating subnetwork, which retrieves the target element value.

## 7  Discussion

As we have shown, neuromodulated RNNs display an increased ability to both perform and generalize on tasks, demonstrating an important computational implication of synaptic gain scaling. This enhanced performance is enabled by the structured flexibility neuromodulation adds to the dynamics of the network, via modulation of the singular values of the low-rank recurrent weight matrix. As we have shown both theoretically and in practice, this flexibility lends itself particularly well to tasks with timing-related variability. In addition, we saw performance gains over the LR-RNN for the multitask paradigm. Curiously, the gating-like dynamics introduced by adding neuromodulation are reminiscent of the canonical LSTM, and we can prove equivalence under certain conditions.

**Limitations.**  One limitation of this work relates to the scale of the networks tested. Our networks were on the scale of $N \approx 100$ neurons at their largest, as opposed to other related works which use neuron counts in the thousands. However, we found that this number of neurons was adequate to perform the tasks we presented. We also have yet to compare our results to neural data, limiting our ability to draw biological conclusions.

**Future Work.**  We are excited at the potential of future work to further bridge the gap between biophysical and recurrent neural network models. To expand on the NM-RNN model, we aim to embrace the broad range of roles neuromodulation can play in neural circuits. Potential future avenues include: (1) sparsifying the rank-1 components of the recurrent weight matrix to better imitate the ability of neuromodulators to act on spatially localized subpopulations of cells; (2) changing the readout function of $\boldsymbol{s}(t)$ to enable it to take both negative and positive values, in line with the ability of neuromodulators to act as both excitatory and inhibitory; and (3) investigating how different neuromodulatory effects may act on different timescales, both during task completion and learning over longer timescales [4, 5]. More generally, each neuron (or synapse) could have an internal state beyond its firing rate which is manipulated by neuromodulators, as in recent work investigating the role of modulation in generating novel dynamical patterns [41, 42]. Beyond neuromodulators, there

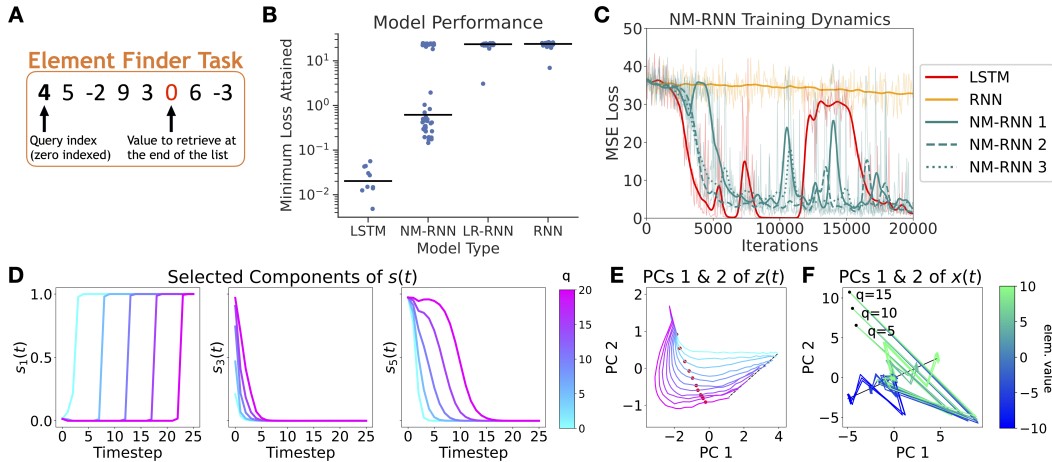

Figure 5: **A.** Visualization of the Element Finder Task. **B.** MSE losses attained across multiple runs in different classes of models trained on the EFT (median is indicated by black lines). **C.** Training loss curves for selected parameter-matched models. The NM-RNNs have hyperparameter counts 1: (M=5, N=18, R=8), 2: (M=5, N=13, R=12), and 3: (M=10, N=12, R=7). **D.** Visualization of selected components of $s(t)$ for an example NM-RNN, shown across different query indices. **E.** Trajectories for the top two PCs of $z(t)$ across different query indices. The different trajectories converge to an approximate line attractor (black) encoding query index. The time at which the queried element arrives is marked in red. **F.** Top two PCs of $x(t)$, visualized for different query indices and target element values. Each trajectory converges to a fixed point on an approximate line attractor encoding element value. Each curve shown in **D**, **E**, and **F** is averaged over 100 trials. Additional visualizations in Supp. Fig. 5-6.

exist a multitude of extrasynaptic signaling mechanisms in the brain, such as neuropeptides and hormones, each with their own computational and modeling implications.

In this work, we only analyzed networks post-training. We are also curious how our computational mechanism of neuromodulation impacts the network during learning. Prior work has modeled the role of neuromodulation in learning, for example, by augmenting the traditional Hebbian learning rule with neuromodulation to implement a three-factor learning rule [43], and by using neuromodulation to create a more biologically plausible learning rule for RNNs [44]. Our neuromodulatory signal could induce similar mechanisms during learning.

## Acknowledgements and Disclosure of Funding

We thank Laura Driscoll, Lea Duncker, Gyu Heo, Bernardo Sabatini, and the members of the Linderman Lab for helpful feedback throughout this project. This work was supported by grants from the NIH BRAIN Initiative (U19NS113201, R01NS131987, & RF1MH133778) and the NSF/NIH CRCNS Program (R01NS130789). J.C.C. is funded by the NSF Graduate Research Fellowship, Stanford Graduate Fellowship, and Stanford Diversifying Academia, Recruiting Excellence (DARE) Fellowship. D.M.Z. is funded by the Wu Tsai Interdisciplinary Postdoctoral Research Fellowship. S.W.L. is supported by fellowships from the Simons Collaboration on the Global Brain, the Alfred P. Sloan Foundation, and the McKnight Foundation. The authors have no competing interests to declare.

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
