# OpenReview forum: "Structured flexibility in recurrent neural networks via neuromodulation"
_NeurIPS.cc/2024/Conference — NeurIPS 2024 poster_

### Official Review · Reviewer_rG4S · 2024-07-12

**Soundness:** 3
**Presentation:** 3
**Contribution:** 3
**Rating:** 6
**Confidence:** 4

**Summary:**

The paper proposes an RNN architecture that include synaptic modulation, motivated by neuromodulatory factors in the brain. In essence, the paper shows it is possible to linearly influence the connectivity matrix of a low-rank RNN by scaling it with the output of smaller RNN, the latter nominally describing neuromodulation. The achieved composition of networks outperforms classical RNNs on tasks with timing and history dependence. The presence of the neuromodulatory signal also allow for multitask learning and reuse of previously learnt dynamics to learn a new task. The paper also compares the performance of the NM-RNN with those of a LSTM, both trained on a digit recall task, where LSTMs are known to outperform RNN. Instead, the result show the added flexibility of NM-RNN.

**Strengths:**

This is a nicely written paper, clear in motivation and generally well-executed. The idea of dynamically scaling the weights of a RNN is a well-reasoned concept, if not entirely novel (as per the prior literature on hypernetworks and synaptic scaling more generally). Adding this form of flexibility could be a useful concept to generalize neural network architectures, and it is certainly an interesting line of study from the perspective of theoretical neuroscience.

**Weaknesses:**

While I generally enjoyed the paper, there are a few issues that impact my initial score.

The domain of impact of the paper is mostly in theoretical neuroscience, since it is not clear from the results that the absolute performance of the NM-RNNs actually exceeds popular constructs such as the LSTM (e.g., F5B). Because a lot of the paper is spent discussing the parallels of the NM-RNN gating mechanism and the LSTM, this becomes important from a mechanistic perspective. Can we gain insight into why there is a plateau in performance, despite the argued parallels in architecture? This could be useful from a fundamental ML perspective.

Related to the above, it would be good to understand actual performance of these networks, in addition to the loss (since there is not always a 1-1 correspondence between the loss and performance).

It is also important to discuss the computational implications of the architecture from a trainability standpoint (see Question below).

From the theoretical neuroscience standpoint, the assumption of neuromodulation acting in this way (weighting low rank components) seems quite abstract and strong. I can understand the argument that networks are low rank, but the idea that modulation effectively selects or weights these components would seemingly require a high degree of precision and some ‘knowledge’ of these components at the level of the modulation generation (I note some discussion of this in the Future Work). Is this realistic, and/or are there ways that the developed models could be substantiated in actual experiments? Are there other ways of neuromodulation entering the model that would be similar effective, or does it really need to be as this sum of low rank components?

**Questions:**

The architecture seems to require an a prior specification of the maximum rank of the connectivity matrix. From an application standpoint, how would this be determined?

Is the proposed method amenable to batch training? Can the authors characterize the trainability/computational implementation considerations of the proposed architecture, relative to say an LSTM? Certainly, there would seem to be more overhead relative to a vanilla RNN.

**Limitations:**

The discussion contains several limitations and future work directions, which are largely appropriate and appreciated.

---

> ### Author Rebuttal · Authors · 2024-08-06
>
> Thank you very much for your helpful comments. We have addressed some shared concerns regarding performance comparisons to LSTMs and vanilla RNNs in the general rebuttal. To respond to your comments regarding weaknesses of the paper:
>
> In general, we don’t believe the NM-RNN will outperform the LSTM. As you mentioned, there is a plateau in performance despite similarities in structure. While our NM-RNN captures some aspects of the LSTM via its synaptic weight scaling, other key features of the LSTM are not implemented. In particular, the input and output weights of the NM-RNN are held constant. Perhaps modulating these weights would help recover more LSTM-like performance. However, our main goal was to offer this model as an alternative to commonly-used low-rank RNNs, as a tool to model biological task completion with neuromodulation. In contrast, LSTMs are not commonly used as “brain-like” models of neural dynamics.
>
> We give example outputs of each model on the Measure-Wait-Go task to highlight specific performance (paper fig. 3D and rebuttal fig. B). For the multitask setting and EFT, example outputs are noisier and are less visibly distinguishable between model types, so we did not visualize them, instead choosing to summarize performance via % correct and loss metrics.
>
> We chose to have neuromodulation differentially impact low-rank components to mimic the selective way neuromodulators act on particular synapses [1], and to situate our model alongside the widely used low-rank RNN. However, this choice is certainly not the only way to model neuromodulation in an RNN—one alternative might be to implement neuromodulatory weights on a mixture of sparse weight matrices, instead of low-rank components. As another example, in Tsuda et al. the authors choose particular subsets of weights on which to apply neuromodulation. However, these subsets are prespecified instead of learned during training.
>
> To answer your questions:
>
> The architecture seems to require a prior specification of the maximum rank of the connectivity matrix. From an application standpoint, how would this be determined?
>
> The rank of the connectivity matrix is indeed a hyperparameter that must be determined before training. In general, we performed a parameter sweep over various numbers of ranks to determine the lowest rank that achieved satisfactory performance. For the MWG task, we chose rank-1 networks due to their theoretical tractability (and because even at this lowest possible rank, the NM-RNN can solve the task). However, to compare to the low-rank RNN we chose rank-3 NM-RNNs, to match prior work which used rank-3 RNNs to model this task [2].
>
> Is the proposed method amenable to batch training? Can the authors characterize the trainability/computational implementation considerations of the proposed architecture, relative to say an LSTM? Certainly, there would seem to be more overhead relative to a vanilla RNN.
>
> This method is amenable to batch training, which we used in the multitask and EFT settings. For the MWG task, the space of possible samples was small enough to do full batch training. In terms of computational implementation, all models must be run sequentially, so while there are slight differences in training time the overall scaling rules likely do not change. We will add a discussion of computational implications to the final version.
>
> Citations
>
> [1] Peter Dayan. Twenty-five lessons from computational neuromodulation. Neuron, 76(1):240–256, 2012.
>
> [2] Manuel Beiran, Nicolas Meirhaeghe, Hansem Sohn, Mehrdad Jazayeri, and Srdjan Ostojic. Parametric control of flexible timing through low-dimensional neural manifolds. Neuron, 111(5):739–753, 2023.

---

> > ### Comment · Reviewer_rG4S · 2024-08-12
> >
> > I thank the reviewers for the rebuttal.  While most of my points have been addressed, I still have a few remaining concerns, namely:
> > - the implementation of transfer learning is still opaque from my perspective. "we froze all recurrent and output weights and retrained only the weights which directly receive the input" is unclear; unless the input of the modulating network is fixed, but not the task-solving network, my impression is that the setup may lead to substantial overwriting/forgetting, and providing information on re-test of prior tasks is crucial in this regard.
> > - Re: batch training. The differences in training time should be described transparently, esp. in the context of the ML impacts of this work.
> >
> > Overall, I am comfortable with my existing score.

---

> > > ### Author Response · Authors · 2024-08-14
> > >
> > > Thank you for your response. We would like to address your remaining concerns:
> > >
> > > To retrain our networks on the new task, we retrained weights which we viewed as "processing" the input for each model. For the NM-RNN, we retrained the parameters of the neuromodulatory subnetwork (input, recurrent, and output weights) and froze the parameters of the larger output-generating subnetwork (input, low-rank recurrent component, and output weights). In essence, we learned a new neuromodulatory signal s(t) to use with existing low-rank recurrent weights in the output-generating subnetwork. We then sought to come up with a fair comparison to the other models (RNNs/LSTMs). Since these models do not have a separate neuromodulatory subnetwork, allowing them to retrain all of their weights seemed like a strong baseline. Instead, we retrained the input weight matrices for the low-rank/vanilla RNNs and LSTMs. In this case the models have to produce new task outputs using fixed recurrent weight matrices (like the NM-RNN) and produce different behaviors via the input weights.
> > >
> > > Our reason for studying this multitask setting was not to test potential for continual learning, but rather to see if the model can solve a new task when it's constrained to re-use prior dynamical motifs. It's true that after retraining there is likely to be overwriting/forgetting in the retrained weights, however we did not attempt to limit this. We think it's interesting that the NM-RNN can solve the new task using low-rank components learned for a prior set of tasks, just weighted differently via the new neuromodulatory signal s(t). Notably, using the previously learned s(t) signals would allow the NM-RNN to immediately switch back to solving old tasks. Although we haven't yet explored whether those signals are maintained or attempted to maintain them, this would be an interesting future direction.
> > >
> > > Re: batch training, thank you for the suggestion. To offer some more concrete details, for the rebuttal figures we trained parameter-matched NM-RNNs (N = 100, M = 20, R = 4) and LSTMs (N = 8)  in the multitask setting using batching (1000 samples for each task, batch size 100). We first trained on the original 3 tasks for 100k iterations of gradient descent, then 50k iterations retraining on the new task. The NM-RNN took about 1.5 hours to train, and the LSTM took about 15 minutes. We believe this discrepancy is due to the large difference in internal state sizes. For instance, the NM-RNN’s neuromodulatory subnetwork alone is 20-dimensional while the highest dimension of any matrix in the LSTM is 8. Additionally, we want to emphasize that we did not focus on optimizing the NM-RNN for speed, but rather wanted to demonstrate its task performance compared to the LSTM. We will ensure these details are made more clear in the final version.

---

### Official Review · Reviewer_znSr · 2024-07-12

**Soundness:** 4
**Presentation:** 3
**Contribution:** 3
**Rating:** 7
**Confidence:** 2

**Summary:**

The authors introduce and implement a novel biologically-inspired variant of standard recurrent neural networks (RNNs), which they evaluate on a number of tasks that require dynamics to generalize across task conditions (Measure-Wait-Go), switch between tasks/task contexts (four-task set from Duncker et al.), and to capture long-term dependencies (Element Finder Task). Their variant, the neuromodulated RNN (NM-RNN), derives from gated coupling between two networks, where the so-called neuromodulation network dynamically scales the recurrent weights of the output-generating network. The authors provide mathematical intuitions about NM-RNNs and how they relate to LSTMs, compare their performance on above-mentioned tasks, and analyse and visualize the networks' learned dynamics.

**Strengths:**

**Originality**: To my knowledge, the architecture is novel.
**Quality**: The paper is very well written and analyses are thorough. The different tasks networks are evaluated on are well-motivated.
**Clarity**: While the paper is well written, clarity could improve if the motivation was spelled out more clearly and if figures were self-contained (via better legends and labels).
**Significance**: The work is an important contribution, but the real-world significance will depend on whether the suggested RNN architecture can be scaled, or used to generate insights about biology (depending on what the author's main motivation is)

**Weaknesses:**

I find the main weakness of the paper to be a lack of clarity in motivation and terminology. I gather the main motivation of the paper is to "bridge the gap between [...] highly biologically-accurate models and general network models (i.e. RNNs) of neuronal activity by adding a biologically-motivated form of structured flexibility" (lines 64-65) - in order to better be able to study flexibility and generalization capabilities observed in biological neural networks.The latter is not very clear from the paper.

The link to neuromodulation features prominently in the Introduction, but should be spelled out more clearly, especially how exactly and strongly it links to the suggested architecture (or it should be clearly stated that the inspiration is only of loose nature). The suggested architecture effectively performs synaptic gain scaling, which is only one of many effects of neuromodulation. Streamlining terminology here might help to improve the clarity of the paper and to guide the reader. As an example, the authors mention dopamine as "a well-known example [...] implicated in motor deficits resulting from Parkinson's disease", but that statement needs to be better connected to the presented study and why it motivates the study.

Both Motivation and Discussion would benefit from an explicit positioning of the approach within the field of NeuroAI: how much does this work contribute to bringing artificial NNs closer to biological NNs, and how much does it help in using ANNs to study BNNs?

In a similar vein, authors may want to discuss their work in relation to:
 - Auzina, I. A., Yıldız, Ç., Magliacane, S., Bethge, M., & Gavves, E. (2023). Modulated Neural ODEs. Proceedings of the 37th Conference on Neural Information Processing Systems. Retrieved from https://github.com/IlzeAmandaA/MoNODE.
- Naumann, L. B., Keijser, J., & Sprekeler, H. (2022). Invariant neural subspaces maintained by feedback modulation. ELife, 11, 76096. https://doi.org/10.7554/eLife.76096

**Questions:**

Conceptual:

Can you clearly define what exactly is meant by **structured flexibility**? Is flexibility in standard RNNs unstructured? Specifically, the EFT analysis and visualization convincingly demonstrate structure in the sense that computations are divided between neuromodulatory and output-generating network; in the Memory/Delay/Anti/Pro, this structure is imposed; but for the MWG task, the authors do not seem to analyse structure.

What have we learned about the computational implications of synaptic gain scaling?

**EFT**

How do the authors predict the performance of the model will change with increasing T? How did the inputs at test time differ from the inputs during training? Is the zeroing out of s(t) components a signature of overparameterization relative to task complexity?


Technical:

Fig. 2A: The corresponding text talks about different components k, but it is not clear what is shown in the figure (a single component across time, or different components? How do changes in s(t) come about?)

Could this be a plug-in replacement for GRUs/LSTMs? If so, how would the model perform standard sequence modeling tasks, such as sequence classification?

The authors don't mention the vanishing gradient issue that RNNs severely suffer from. Did the not have this problem?

Fig 3. E: please offer more interpretation for what is shown.

Minor:

ln. 104: "Likewise, artificial neural networks trained to solve tasks that mimic those found in neural experi-
ments also often exhibit low-rank structure." Please provide a reference.

Fig 4 A: This panel is not clear. What is shown? Please add labels to all traces and to the axes, and make clear how the subpanels’ axes relate to each other.

The Memory/Anti task is not readily understandable from the description and visualization in the paper. Please explain this better to make the paper self-contained.

Fig 5C what are the squiggly lines? Fig E not clear where trajectories begin and end, black line not visible. Fig F similarly hard to see/interpret.

Eq. 4: Please specify what \sigma is.

Fig 3 references mixed up in ln.208.

ln. 267: "Curiously, the positive/negative relationship flips for θ ∈ (π, 2π), likely relating to the symmetric structure of sine and cosine." where can we see this?

**Limitations:**

The authors adequately address the limitations of the work.

---

> ### Author Rebuttal · Authors · 2024-08-06
>
> Thank you very much for your thorough comments. Indeed, our motivation was to create a model somewhere between RNNs and biologically-accurate biophysical models. Specifically, we identified neuromodulation as a feature of biological networks that is not often modeled in RNNs. The goal of this paper is to study the impact of synaptic scaling, one specific effect of neuromodulation, on the performance and generalization of RNN models. We have discussed our motivation further in the General Rebuttal and will make this positioning more clear in the final version.
>
> Thank you for suggesting Auzina et al. and Naumann et al., we will add these papers to our related work section to highlight the use of modulatory signals in non-RNN models.
>
> To address your questions:
>
> Conceptual:
>
> By structured flexibility, we mean the introduction of additional parameters (i.e. flexibility) to a model, with these parameters having a clear motivation and interpretation (i.e. structure). In the case of our NM-RNN, the structured flexibility comes from adding a flexible neuromodulatory signal which impacts RNN dynamics in a constrained way (by scaling synaptic weights). In the EFT and multitask settings, we show that this allows the network to distribute different parts of the task across the subnetworks. In the MWG task, we show that ablating each component of the neuromodulatory signal leads to performance deficits (paper fig. 3F). In particular, ablating s_3(t) destroys the ability of the network to produce output ramps of different slopes, implying that the neuromodulatory subnetwork is involved in controlling the output interval length. In all three ablations, the general ramping shape of the output is preserved, implying that this behavior is stored in the dynamics of the output-generating network.
>
> Overall, we learned that synaptic gain scaling can improve generalization capabilities of low-rank RNNs and make them more like LSTMs. More generally, we offer this model as an alternative to commonly-used low-rank RNNs, as a tool to model biological task completion with neuromodulation.
>
> EFT:
>
> Each of our models was first trained in a continual learning regime in which each step of gradient descent was performed using a randomly generated batch of input sequences. Then, at test time, test input sequences were again sampled i.i.d. from the distribution of all valid EFT sequences. (It should be noted that for main paper figs. 5C, D, and F, we randomly generated test inputs while holding the query index and/or target element value fixed.)
>
> As the sequence length T increases while keeping the NM-RNN model size, number of training batches, and training batch size fixed, we predict that model performance will degrade. The set of possible EFT input sequences grows exponentially in T, making it challenging for the model to solve the task within the same number of training iterations; indeed, this is what we observed in our preliminary simulations. However, we expect that model performance would improve if the number of training iterations and/or the model size is suitably increased.
>
> Technical:
>
> Fig. 2A: The corresponding text talks about different components k, but it is not clear what is shown in the figure (a single component across time, or different components? How do changes in s(t) come about?)
>
> This figure is meant as an illustrative example of how the neuromodulatory signal s(t) can impact dynamical timescales. In this example, s(t) is a 1-dimensional signal so k=1. The changes in s(t) are artificially imposed for illustrative purposes, to show how the decay rate of w(t) is modulated by the value of s(t). We will clarify this in the final version.
>
> Could this be a plug-in replacement for GRUs/LSTMs? If so, how would the model perform standard sequence modeling tasks, such as sequence classification?
>
> We are not arguing to replace GRUs/LSTMs in standard ML workflows. However, we show that they do share some capabilities, achieving LSTM-like performance on some tasks.
>
> The authors don't mention the vanishing gradient issue that RNNs severely suffer from. Did the not have this problem?
>
> This issue did not arise for the tasks that we trained on. To speculate, perhaps the NM-RNN does not encounter this issue as often due to its similarities to the LSTM. This could also be a potential reason why the RNNs are failing at the EFT.
>
> Fig 3E: please offer more interpretation for what is shown.
>
> In fig. 3E we show the 3-channel neuromodulatory signal s(t) during the MWG task. On the left, the signals are aligned to when networks receive the measure cue. On the right, the signals are aligned to when networks receive the wait and go cues. The left figure shows how the s(t) responds to the measure cue and evolves throughout the task. The right panel shows how s(t) readies responses for the different interval lengths. In particular, we can see that between the wait and go cues, s_1(t) separates shorter intervals from longer ones, setting up initial dynamics for the go period when the ramp is generated. The third component s_3(t) seems to be involved with ending the output ramp, since it saturates first for the shorter intervals and then the longer ones.
>
> Minor:
>
> ln 104: we will add a citation to Schuessler et al. “The interplay between randomness and structure during learning in RNNs”, NeurIPS 2020
>
> Fig. 4A: we have updated this figure in the PDF attached to the general rebuttal. Please let us know if additional adjustments would help to clarify.
>
> Fig. 5: The squiggly lines are the actual loss curves, while the bold lines show a moving average to indicate overall trends. We will add improved explanations and better-contrast lines to E and F.
>
> Eq. 4: \sigma is the sigmoid nonlinearity, we will clarify this in the final version.
>
> Fig. 3/ln. 208: thank you for catching this, we will amend it in the final version.
>
> ln 267: we have added these plots to the general rebuttal PDF and will include them in the final paper fig. 4D.

---

> > ### Comment · Reviewer_znSr · 2024-08-11
> >
> > I thank the authors for their detailed reply to my review and the improved figures. I'd like to ask the authors to include their clarifications in the final version of the paper as they deem helpful for future readers; especially the positioning within the field of NeuroAI is important. I support acceptance of the paper.

---

### Official Review · Reviewer_Eb4N · 2024-07-13

**Soundness:** 3
**Presentation:** 3
**Contribution:** 2
**Rating:** 6
**Confidence:** 4

**Summary:**

This work studies the effects of synaptic gain scaling, a neuromodulatory mechanism, on the performance of task-trained low-rank RNNs – which have been used to understand the dynamics and other finer details of neural computation. Specifically, it introduces a simple time-varying neuromodulatory mechanism implemented as a hypernetwork to modulate the weights of low-rank RNNs trained on multiple tasks. The authors draw connections between the proposed model and LSTMs by theoretically demonstrating that the neuromodulation provides for a form of gating. Finally, the authors show that the proposed model outperforms low-rank RNNs and is comparable to LSTMs in multi-task settings and tasks involving long-term dependencies.

**Strengths:**

1. The writing is clear and the overall presentation of the paper is good – related work is adequately described, and the model formulation is clear and simple but effective.
2. The authors theoretically study the links between the proposed NM-RNNs and LSTMs and show that, under certain assumptions, NM-RNNs learn gating mechanisms similar to LSTMs leading to their improved performance over LR-RNNs on certain tasks. To my knowledge, this theoretical analysis is novel and sound.
3. The authors perform several clearly motivated numerical experiments to analyze the computational implications of the proposed model. Particularly, they demonstrate that their model outperforms ordinary LR-RNNs in multi-task settings and capturing long-term dependencies.

**Weaknesses:**

1. The model is limited in its bio-realism, so it is unclear how well this would match to computations performed in the brain. Specifically, there are no testable predictions for this model of neuromodulation and no comparison with neural data (as acknowledged by the authors in the limitations), so this limits the significance of the work.
2. The model does not seem to perform as well as LSTMs in the long-term dependencies task (the loss attained by LSTMs is far lower). Furthermore, the authors do not show comparisons with LSTMs and vanilla RNNs for the timing task and the multi-task setting. It would be important to show how the NM-RNN performs in comparison to at least the LSTMs as well in these settings.
3. On a related note to the previous point, the authors have not compared their model to existing implementations of neuromodulation-infused RNNs. Given the similarity of this work to Tsuda et al., 2021 [1] (which also seems to explain observations in neural data from Drosophila), is it possible to compare the NM-RNN to this model, and perhaps other approaches such as those introduced by Liu et al., 2022 [2]?
4. Most importantly, it seems like the scalability of this model is not well-studied and this is also related to the fact that the tasks considered are very simplistic. While these simple tasks allow for some interpretability, I think it would also be important to benchmark the NM-RNN on more complex tasks. For example, studying how it adapts under perturbation in a reaching/control task – this would be a more complex task and could also help understand the links between neuromodulation and learning. In general, moving beyond toy-ish tasks would strongly improve the experimental section of the paper.

(References provided in Questions section.)

**Questions:**

See the Weaknesses section. Some additional questions:
1. Have the authors tried training the neuromodulatory subnetwork alone while treating the task subnetwork as a reservoir? This could potentially provide for a parameter-efficient multi-task learning framework (loosely related, see the recent paper by Williams et al., 2024 [3]).
2. Could the authors clarify what the three different NM-RNN curves are in Fig. 5C? If they are different seeds or configurations, this should be specified.
3. Have the authors tried using other activation functions in calculating $\mathbf{s}(\mathbf{z}(t))$, i.e., not restricting the neuromodulatory signal to values between 0 and 1 (and allowing negative values)? Could this lead to improved performance?

**References:**
1. Tsuda et al. "Neuromodulators generate multiple context-relevant behaviors in a recurrent neural network by shifting activity hypertubes." bioRxiv (2021): 2021-05.
2. Liu et al. "Biologically-plausible backpropagation through arbitrary timespans via local neuromodulators." Advances in neural information processing systems 35 (2022): 17528-17542.
3. Williams et al. "Expressivity of Neural Networks with Random Weights and Learned Biases." arXiv preprint arXiv:2407.00957 (2024).

**Rebuttal update:** Confidence increased from 3 to 4. I am in favor of accepting the paper.

**Limitations:**

The limitations have been adequately discussed in the paper (specifically, biological realism, comparison to neural data, and scalability).

---

> ### Author Rebuttal · Authors · 2024-08-06
>
> Thank you very much for your feedback. We have addressed some of your comments in the general rebuttal, in particular the second point under Weaknesses (regarding performance comparison to LSTMs and vanilla RNNs). We would also like to respond to the additional weaknesses noted.
>
> While we currently have no comparison with neural data, we believe this model can offer testable predictions of how diseases involving neuromodulator deficiencies and dysfunction impact movement and timing. As RNNs have been used to generate testable predictions of neural function, we believe our model offers a biologically meaningful extension by scaling synaptic weights as biological neuromodulation has been shown to do.
>
> We did not compare our work with the prior methods mentioned in the Related Work section because of a few key differences in motivation and implementation. The work of Tsuda et al., while similar, requires prespecification of (1) which neurons are neuromodulated and (2) the constant value that the neuromodulation takes. Our paradigm allows both of these parameters to be learned by the model. It was unclear how to set the aforementioned parameters in the Tsuda model in order to make a comparison. While Liu et al. also studied the impact of neuromodulation in RNN models, their work focused primarily on the impact of neuromodulation on training and not during task performance. Due to the difference in motivation, we did not compare to their model.
>
> We agree that training our model on more complex tasks will be an important step forward. Since vanilla RNNs are used to model more complex tasks and combinations of tasks [Driscoll], we believe our model should also be amenable to training in these situations. We appreciate the suggestion for studying reaching/control perturbation tasks and agree that training the NM-RNN in this task setup could help provide valuable insights into the relationship between neuromodulation and adaptation.
>
> To answer your questions:
>
> Have the authors tried training the neuromodulatory subnetwork alone while treating the task subnetwork as a reservoir? This could potentially provide for a parameter-efficient multi-task learning framework (loosely related, see the recent paper by Williams et al., 2024 [3]).
>
> We like this idea and think it would be an interesting extension for the multitask section of the paper. While we did not try random reservoir dynamics as in Williams et al., we did analyze the effect of retraining only the neuromodulatory signal while leaving the low-rank dynamical components fixed when learning a new task (paper section 5). These results could be viewed as a proof-of-concept that the NM-RNN can effectively reuse existing dynamics when learning new tasks.
>
> Could the authors clarify what the three different NM-RNN curves are in Fig. 5C? If they are different seeds or configurations, this should be specified.
>
> We apologize for the confusion—these are different hyperparameter settings for the NM-RNN, parameter-matched to the N=10 LSTM. We will update the caption of this figure in the final version to indicate the exact hyperparameters used in each curve. For reference the settings are (M=5, N=18, R=8), (M=5, N=13, R=12), and (M=10, N=12, R=7). Our overall goal in this figure was to show that LSTMs and NM-RNNs are able to solve the task, while vanilla RNNs cannot train on it.
>
> Have the authors tried using other activation functions in calculating z(t), i.e., not restricting the neuromodulatory signal to values between 0 and 1 (and allowing negative values)? Could this lead to improved performance?
>
> While this may lead to improved performance, we believe it would negatively impact the biological plausibility of the modulatory signal. We capped the neuromodulatory signal at 1 to model a saturating effect of neuromodulators on downstream synaptic strength—the assumption being that synaptic strength cannot increase indefinitely. Similarly, we did not allow negative saturation, instead setting the floor value of the neuromodulator to 0 to indicate complete silencing of synaptic connections and preserve the sign of the connection.

---

> > ### Comment · Reviewer_Eb4N · 2024-08-10
> >
> > I thank the authors for the rebuttal and appreciate the additional simulations with LSTMs. The authors' reasoning behind not comparing their method with Tsuda et al. and Liu et al. sounds reasonable to me. I am mostly satisfied with the other responses, and while I'd have liked to see more complex tasks/multi-task experiments, I completely understand not being able to do so due to the limited time available, and this is not a reason for me to reject the paper.
> >
> > Overall, I quite like the straightforwardness of this paper, the theoretical results connecting the proposed mechanism to LSTMs, and the general presentation, figures and interpretable experiments. I maintain my positive opinion and score, and I think this will be a good contribution to computational neuroscience and the audience at NeurIPS.

---

### Official Review · Reviewer_JBgz · 2024-07-13

**Soundness:** 2
**Presentation:** 2
**Contribution:** 3
**Rating:** 5
**Confidence:** 4

**Summary:**

This work mimic the synaptic plasticity observed in brain which is driven by neuromodulators to develop neuro-inspired artificial neural networks. It proposes the neuromodulated NM-RNN, it has a neuromodulatory subnetwork that outputs a low-dim output that will scale the synaptic weights of low-rank RNN. It has connection to LSTMs with similar gating mechanisms, or even the dynamics of NM-RNN could be reproduced with a LSTM. The model is better capturing long-term dependencies. The work also demonstrate how this framework is applicable in multitask learning.

**Strengths:**

**Methods**

This work focused on an important question to incorporate nonstationarity observed in biological network into artificial network. It implemented the potential mechanisms that synaptic plasticity controlled by the neural modulators. The subnetwork to outputs modulatory inputs for the scaling the low-rank RNN's weights is a novel design.

**Evaluation**

The NM-RNN has been benchmarked with multiple classic artificial neural networks (RNN, LSTM) on multiple tasks including measure-wait-go, multitask learning, element finder task to demonstrate its capabilities of capturing long-term dependencies.

**Weaknesses:**

**Method**

1. The design choice of rank-1 in modulatory subnetwork should be described, more hyperparameters should be explored.

**Baselines**

1. The proposed NM-RNN models has multiple connections with LSTM, the novelty is therefore limited. Meanwhile, its performance is not comparable to LSTM as shown in Fig5 B.
2. The baselines are limited, i.e. comparisons with transformers which also has input-dependent attention weights which also model the nonstationarity, and relevant to neural plasticity.

***Evaluation**

1. The tasks and  LSTM is not evaluated in measure-wait-go, multitask learning tasks. Adding more tasks to demonstrate the effectiveness of NM-RNN to captures long-term dependencies.

**Questions:**

1. Why choose rank-1 for modulatory subnetwork?

2. Why the training dynamics (MSE loss) for all models are noisy and fluctuated, any optimization techniques (regularization, normalization, scheduler might help with the training dynamics), how to guarantee the convergence of the model?

3. Any computational efficiency that NM-RNN might bring in?

**Limitations:**

No potential negative societal impact. The limited scale of the network has been discussed.

---

> ### Author Rebuttal · Authors · 2024-08-06
>
> Thank you very much for your feedback. We have addressed some of your comments from the Weaknesses section in the general rebuttal (in particular, concerns about Evaluation and Baselines). We would also like to clarify why we did not initially provide an LSTM baseline for the Measure-Wait-Go and multitask settings. The goal of this project is to offer a new model relevant to the field of computational neuroscience. Low-rank RNNs are commonly used tools in computational neuroscience due to their low-dimensional internal dynamics. We augmented the low-rank RNN with an interpretable modulatory parameter, and showed the comparative effectiveness of low-rank RNNs and NM-RNNs on tasks related to neuroscience. We also showed that NM-RNNs help to recover some of the performance gains of LSTMs. However, LSTMs are not commonly used as “brain-like” models when studying neural population dynamics. One of our contributions was to show the similarities between our more “brain-like” NM-RNN and the popular LSTM. Like LSTMs, transformers are not typically used to model neural population dynamics, so we did not include comparisons to these models.
>
> To address your questions:
>
> Why choose rank-1 for modulatory subnetwork?
>
> To clarify, the modulatory subnetwork is full-rank and the output-generating subnetwork is low-rank. In our study of the Measure-Wait-Go task we chose to present results with rank-1 and rank-3 NM-RNNs. We analyzed rank-1 networks due to their theoretical tractability, and rank-3 networks in order to compare to existing work which used rank-3 low-rank RNNs on this task [1]. For the multitask setting, we chose rank-3 NM-RNNs by sweeping over ranks and determining the rank required to consistently train on the first three tasks. For the element-finder setting we tried many combinations of rank-size parameters to compare to the LSTM.
>
> Why the training dynamics (MSE loss) for all models are noisy and fluctuated, any optimization techniques (regularization, normalization, scheduler might help with the training dynamics), how to guarantee the convergence of the model?
>
> We agree that the training dynamics for the Element-Finder task (EFT) are quite noisy, and would likely become smoother with the use of regularization techniques. However, our main point with this figure was to show general trends, i.e. vanilla RNNs struggle to solve EFT while some LSTMs and NM-RNNs are able to learn it consistently. In addition, we did not show the training dynamics for any of the other tasks, but can share empirically that they were much smoother than in the EFT, suggesting that the noisy learning may also be due to the relative complexity of this task.
>
> Any computational efficiency that NM-RNN might bring in?
>
> All models are run sequentially, so while there is a slight difference in computational efficiency between each individual step, the scaling laws likely don’t change. In general, you raise an interesting point that the matrix multiplication cost for running a step of a low-rank RNN is smaller than that of a vanilla RNN. So, low-rank RNNs and NM-RNNs will be faster to evaluate compared to a vanilla RNN with similar neuron count. We will add a discussion of computational implications to the final version.
>
> Citations
> [1] Manuel Beiran, Nicolas Meirhaeghe, Hansem Sohn, Mehrdad Jazayeri, and Srdjan Ostojic. Parametric control of flexible timing through low-dimensional neural manifolds. Neuron, 111(5):739–753, 2023.

---

> ### Comment · Reviewer_JBgz · 2024-08-13
>
> Thanks for the authors' response. While I am not convinced that LSTM and transformers are not "brain-like", while RNNs are. I view them all more as predictive models instead of mechanistic models (i.e. hodgkin huxley model). And a lot of recent works have started to use transformers model neural population dynamics [1][2][3], and outperforms RNNs. I think adding more comparisons with baselines, and improving the optimization will be helpful. I decided to maintain my original score.
>
> [1] Representation learning for neural population activity with Neural Data Transformers.
>
> [2] STNDT: Modeling Neural Population Activity with a Spatiotemporal Transformer.
>
> [3] Neuroformer: Multimodal and Multitask Generative Pretraining for Brain Data.

---

> ### Comment · Reviewer_Eb4N · 2024-08-14
>
> I just wanted to step in and provide my perspective here as another reviewer, because I believe that the papers this reviewer mentions are orthogonal to what I believe is the goal of this paper.
>
> The goal here is to propose a neuromodulatory mechanism inspired by synaptic gain scaling, as a circuit-level mechanistic model of how modulatory signals could reconfigure the dynamics of a recurrent network to perform different tasks. I view it as a mechanistic model rather than a predictive one, i.e., it describes how a mechanism that is computationally similar to what is observed in the brain enables multi-task flexibility, rather than serving as a better predictive model or advancing performance from a deep learning perspective. In short, it is trying to computationally analyze the effect of synaptic gain scaling on multi-task performance.
>
> Several works have used recurrent neural networks to describe how the brain could be performing certain task-related computations, including influential work by Yang et al. [1] and Driscoll et al. [2]. While I'm not arguing here that somehow vanilla RNNs are more bio-plausible or better brain models than LSTMs, I believe there are different levels of modeling and biological realism in the computational neuroscience literature. Circuit-level computational models such as this work eschew synapse-level biological plausibility and aim to study computations through the lens of dynamics [3], task performance, or other hallmarks of neural computation like flexibility and generalization.
>
> I would thus argue that RNNs are good and simple/minimal models to use here – we are aware of recurrent circuits in various brain regions [4,5,6,7], but there is no similar support for transformer-like architectures or self-attention, to my knowledge (although on the other hand, these models are, recently, used at the cognitive level of modeling [8,9]). And given that the kind of tasks being performed here are related to working memory, which relies on the prefrontal cortex [10], which is further known to contain several recurrent microcircuits [6,7], I'd argue that RNNs are better computational models to use here than transformers if the goal is to model neural computation (which I strongly believe to be the case here).
>
> Finally, and perhaps most importantly, the papers that the reviewer refers to here are not trying to model how the brain could compute, or how mechanisms in the brain could enable multi-task flexibility. Works such as [11,12,13], using transformers to model neural population activity, are harnessing advances in deep learning to serve as better representation learning methods or predictive models of behavior from neural activity, for e.g., towards the goal of building better decoders from brain-computer interfaces – they are not meant to serve as models of how the brain could compute.
>
> Overall, I just wanted to provide my views here and I hope to engage in further discussion with this reviewer and/or the authors on this.
>
> **References:**
> 1. Yang, Guangyu Robert et al. “Task representations in neural networks trained to perform many cognitive tasks.” Nature neuroscience vol. 22,2 (2019): 297-306.
> 2. Driscoll, Laura N et al. “Flexible multitask computation in recurrent networks utilizes shared dynamical motifs.” Nature neuroscience vol. 27,7 (2024): 1349-1363.
> 3. Driscoll, Laura N et al. “Computation through Cortical Dynamics.” Neuron vol. 98,5 (2018): 873-875.
> 4. Douglas, Rodney J, and Kevan A C Martin. “Recurrent neuronal circuits in the neocortex.” Current biology : CB vol. 17,13 (2007): R496-500.
> 5. Wang, Xiao-Jing. “Decision making in recurrent neuronal circuits.” Neuron vol. 60,2 (2008): 215-34.
> 6. Mante, Valerio et al. “Context-dependent computation by recurrent dynamics in prefrontal cortex.” Nature vol. 503,7474 (2013): 78-84.
> 7. Fuster, J M. “Memory networks in the prefrontal cortex.” Progress in brain research vol. 122 (2000): 309-16.
> 8. Didolkar, Aniket, et al. "Metacognitive Capabilities of LLMs: An Exploration in Mathematical Problem Solving." arXiv preprint arXiv:2405.12205 (2024).
> 9. Webb, Taylor, et al. "A Prefrontal Cortex-inspired Architecture for Planning in Large Language Models." arXiv preprint arXiv:2310.00194 (2023).
> 10. Funahashi, Shintaro. “Working Memory in the Prefrontal Cortex.” Brain sciences vol. 7,5 49 (2017).
> 11. Ye, Joel, and Chethan Pandarinath. "Representation learning for neural population activity with Neural Data Transformers." arXiv preprint arXiv:2108.01210 (2021).
> 12. Le, Trung, and Eli Shlizerman. "Stndt: Modeling neural population activity with spatiotemporal transformers." Advances in Neural Information Processing Systems 35 (2022): 17926-17939.
> 13. Antoniades, Antonis, et al. "Neuroformer: Multimodal and multitask generative pretraining for brain data." arXiv preprint arXiv:2311.00136 (2023).

---

### Author Rebuttal · Authors · 2024-08-06

We would first like to thank all of the reviewers for their insightful and thorough comments on our paper. We have carefully considered your feedback and appreciate your help in both strengthening our current paper’s claims and providing future directions for us to consider.

In this general rebuttal, we would like to provide answers to some common concerns as well as highlight the rebuttal figures shown in the attached PDF. First, a request shared among multiple reviewers was to compare the performance of our NM-RNN to LSTMs and vanilla RNNs in the Measure-Wait-Go (MWG) task and multitask setting. The comparisons among all four models (parameter-matched low-rank RNNs (R=3, N=106), NM-RNNs (R=3, M=5, N=100), vanilla RNNs (N=31), and LSTMs (N=15)) on the MWG task are shown in fig. A&B. In fig. A we’ve plotted the loss for 10 instances of each model type, both on linear and log scales. On trained intervals, the LSTM achieves the lowest median L2 loss due to its ability to accurately shape the output ramps (fig. B). However, on the extrapolated intervals the NM-RNN achieves the lowest median loss out of all four model types. Looking at the example outputs in fig. B, we see that this is because the NM-RNN better generalizes the slope of the output ramp to extrapolated intervals, compared to the LSTM. Overall, we see that our NM-RNN enjoys enough complexity to both train and generalize well on the task, without suffering from generalization performance losses like the LSTM.

We have also trained all four model types (parameter-matched low-rank RNNs (R=3, N=100), NM-RNNs (R=3, M=20, N=100), vanilla RNNs (N=18), and LSTMs (N=8)) in the multitask setting, on both the three initial and one retrained tasks. For retraining the LSTM and vanilla RNN, we froze all recurrent and output weights and retrained only the weights which directly receive the input. The percent correct metric for 10 instances of each model is shown in fig. D. On the trained tasks, the NM-RNNs, vanilla RNNs, and LSTMs are able to train to high accuracy. On the retrained task, these three models still show similar high performance. While there is an outlier NM-RNN (also shown in the paper), the majority of the NM-RNNs achieve similar retraining performance to both LSTMs and vanilla RNNs.

We would also like to use this general rebuttal to clarify our goals. In particular, our motivation was not to create a model that would outperform the LSTM on most tasks; rather, we aimed to create a version of an RNN which includes a biologically motivated implementation of synaptic gain scaling. One of the many known effects of neuromodulation on neural computation is synaptic scaling. This work aims to highlight how traditional low-rank RNN models of task completion may be overlooking synaptic scaling, a relevant biological factor that we show aids in performance and generalization. In developing this model we discovered similarities between our low-rank modulation and gating in LSTMs. This prompted us to further explore the relationship between the NM-RNN and LSTM. However, LSTMs are not commonly used as “brain-like” models in computational neuroscience, so we did not initially compare their performance on the neuroscience-relevant tasks. We believe this model offers a unique framework by which to study and hypothesize the impact of neuromodulation on task performance and generalization in biological networks. For example, we could use NM-RNNs trained on timing tasks to predict the effect of decreased dopamine.

Rebuttal figs. C and E are both in response to Reviewer znSr. Fig. C is an updated version of paper fig. 4A. We aimed to clarify the distinctions between the four tasks in the multitask setting by adding highlights to the figure indicating separate task labels, inputs, and outputs, as well as verbal description of task goals and boxes around the separate tasks. Fig. E shows the flip in sign of PCs 4&5 of our example multitask NM-RNN. The network distinguishes between Pro/Anti versions of the same task by flipping the sign of PC5 as the measured angle crosses $\pi$.

---

### Decision · Program_Chairs · 2024-09-25

**Decision:**

Accept (poster)

**Comment:**

This paper proposes a low-rank recurrent neural network model with neuromodulation that modulates the strength of the low-rank factors, and it develops analytical connections with LSTMs. Overall, the reviewers agreed this paper presents a novel solution to an important problem in theoretical neuroscience, while establishing theoretically sound connections with LSTMs. Therefore, I recommend acceptance.